# A valve-like mechanism controls desensitization of functional mammalian isoforms of acid-sensing ion channels

**Yangyu Wu, Zhuyuan Chen, Cecilia M Canessa***

Department of Basic Sciences, Tsinghua University School of Medicine, Beijing, China

**Abstract** ASICs are proton-gated sodium channels expressed in neurons. Structures of chicken ASIC1 in three conformations have advanced understanding of proton-mediated gating; however, a molecular mechanism describing desensitization from open and pre-open states (steady-state desensitization or SSD) remains elusive. A distinct feature of the desensitized state is an 180° rotation of residues L415 and N416 in the β11- β12 linker that was proposed to mediate desensitization; whether and how it translates into desensitization has not been explored yet. Using electrophysiological measurements of injected *Xenopus* oocytes, we show that Q276 in β9 strand works with L415 and N416 to mediate both types of desensitization in ASIC1a, ASIC2a and ASIC3. Q276 functions as a valve that enables or restricts rotation of L415 and N416 to keep the linker compressed, its relaxation lengthens openings and leads to sustained currents. At low proton concentrations, the proposed mechanism working in only one of three subunits of the channel is sufficient to induce SSD.
DOI: https://doi.org/10.7554/eLife.45851.001

*For correspondence:
cecilia.canessa@yale.edu

Competing interests: The authors declare that no competing interests exist.

## Introduction

ASICs are sodium channels gated by external protons expressed in neurons. In the peripheral nervous system activation of ASICs induces pain (*Yu et al., 2010*) while in the central nervous system modulates synaptic plasticity influencing the generation of memories (*Wemmie et al., 2002*; *Yu et al., 2018*), fear conditioning (*Coryell et al., 2007*; *Wemmie et al., 2003*) and extinction (*Wang et al., 2018*). Pathological activation of ASICs contributes to neuronal damage induced by ischemia (*Duan et al., 2011*; *Xiong et al., 2004*) and inflammatory processes (*Friese et al., 2007*). A characteristic of human ASICs is that protons act not only as agonists but also as strong inhibitors by inducing two types of desensitization: low-pH desensitization shuts channels from the open state, and steady-state desensitization (SSD) shuts channels from pre-open closed states (*Waldmann et al., 1997*). SSD in hASIC1a is almost complete at pH 7.0, whereas maximal activation requires pH <6.5. Desensitization shapes the time course of currents in distinct ways in each of the three functional ASIC isoforms and is important to limit depolarization of neurons under global states of acidosis or local ischemia. Although mutations in many domains of the protein alter desensitization (*Bonifacio et al., 2014*; *Li et al., 2010a*; *Li et al., 2010b*; *Liechti et al., 2010*; *Roy et al., 2013*; *Vullo et al., 2017*), there is still incomplete knowledge of the molecular mechanisms underlying this process and how it controls channel function. The atomic structures of chicken ASIC1 in closed (*Yoder et al., 2018*), open (*Baconguis et al., 2014*), and low-pH desensitized states (*Gonzales et al., 2009*) have paved the way to address the molecular underpinning of desensitization. A distinct feature of the low-pH desensitized channel is a swap of side chain orientation of L414 and N415 located in the β11-β12 linker (*Baconguis and Gouaux, 2012*) that led Gouaux et al. to hypothesize that the linker uncouples the upper ECD from the lower channel – it works as a

molecular clutch (*Yoder et al., 2018*) - allowing the pore to adopt the non-conducting conformation of the desensitized state.

Here, we used human ASIC1a to test such hypothesis and shed further light on the molecular basis underlying desensitization. We found that the highly conserved residue Q276, located in the middle of the β9 strand works together with L415 and N416 to desensitize ASIC1a, ASIC2a and ASIC3. We propose a valve-like mechanism, named here Q valve in reference to the central role of Q276, wherein rotation of L415 and N416 and compression of the β11-β12 linker are locked by Q276 stabilizing the desensitized state and ensuring both complete SSD and low-pH desensitization. Substitutions of any of the three essential components result in a leaky valve leading to incomplete or no desensitization.

## Results

### Q276 is a key residue involved in desensitization of human ASIC1a

Q276 is located midway β9 strand in the palm of hASIC1a with the side chain facing the interior of the central vestibule (*Jasti et al., 2007*); it is conserved in the four mammalian ASIC isoforms and in other vertebrate species sequenced to date (*Figure 1A*). Mutations in this position changed channel desensitization kinetics. All substitutions generated functional channels with currents averaging $22 \pm 6$ µA/cell with the exception of Q276D that had small currents ($1.5 \pm 1.1$ µA/cell); all exhibited either slow desensitization rates or various magnitudes of sustained current that did not desensitize completely. *Figure 1B* shows normalized current traces of a few representative mutants superimposed on wild-type channels. The dashed vertical line marks the time corresponding to three times the desensitization τ value of wild-type channels ($\tau_{wt}1.8$ s). All mutants remained partially open at $x3\tau_{wt}$; Q276E currents eventually desensitized completely though very slowly (τ 7.2 s), whereas Q276G was remarkable because desensitization was mostly abolished. Additional representative examples of other mutants are shown in *Figure 1—figure supplement 1*. Hydrophobicity or charged groups did not play a main role rather size of the side chain seemed to matter namely, small and large side chains favored sustained currents whereas intermediate size -similar to glutamine- more completely desensitized. Relation of the fraction of remaining current after $x3\tau_{wt}$ and the volume of the corresponding side chain ($\text{Å}^3$) (*Esque et al., 2010*) is shown as a filled circle on each bar presented in *Figure 1C*. Decrease or increase of size away from Q (black circle) increased the magnitude of sustained currents and/or slowed desensitization rates.

Mutations of residue Q278 (Q278N or Q278G) that is located two positions down Q276 in the same β9 strand (*Figure 1A* yellow sticks), with side chain also facing the interior of the central vestibule, produced currents with properties indistinguishable from those of wild type (*Figure 1—figure supplement 1*), thus not all positions in the β9 strand alter channel function implying that the kinetic changes are specifically induced by substitutions of Q276.

### Substitutions of Q276 lengthen the duration of channel openings but do not alter conductance or ion selectivity

In outside-out patches we examined single channels of Q276G and Q276N that exhibit mainly sustained current or peak and sustained currents, respectively. Most patches contained many channels (*Figure 2A*) only a few (8 out of 30) had one to three channels enabling detection of individual opening events. These lasted many seconds, in some instances up to 80 s (*Figure 2B–C*). A histogram of the duration of individual events to determine the exact value of the mean open time was not feasible owing to the few observed events, some sweeps showed only 3 to 8 opening events during 3-min recordings at pH 6.5. Nevertheless, the duration of mutant openings was markedly longer than that of wild-type channels. For comparison *Figure 2D* shows three successive weeps of a patch expressing wild-type hASIC1a containing three channels. Openings are brief ranging from 4 to 230 ms with a mean of $83.5 \pm 99$ ms in the example, and are followed by desensitization as others and our group have previously reported (*Li et al., 2010b*; *Sutherland et al., 2001*). Long openings arise from a decrease in the transition rate from the open to desensitized state that is mutant channels do not enter the desensitized state as readily as wild-type channels. Openings were either uninterrupted long-lived open states or interrupted by transient closures. These closures were too brief to represent desensitized states, suggesting that desensitized channels rapidly return either to the

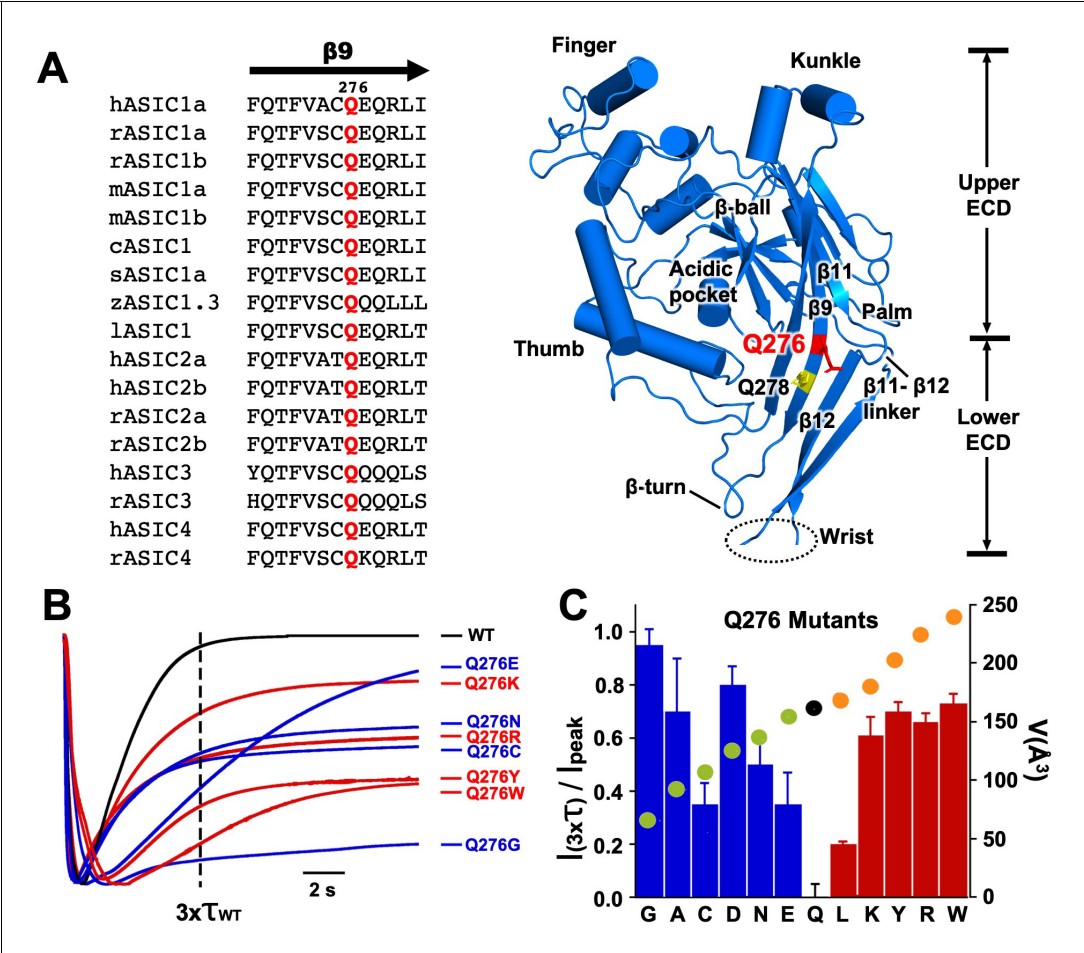

**Figure 1.** Substitutions of residue Q276 diminish or abolish desensitization. (**A**) Sequence alignment of β9 strand of ASIC isoforms from various vertebrate species. Ribbon representation of the crystal structure of the extracellular domain (ECD) of a single subunit of chicken ASIC1. Positions of Q276 and Q278 in β9 are shown as sticks colored red and yellow, respectively. The β11- β12 linker near Q276 has been proposed to functionally divide the ECD into Upper ECD and Lower ECD. (**B**) Superimposed normalized currents of wild type hASIC1a and Q276 mutants activated by pH 6.5. The vertical dashed line marks the time corresponding to 3x the time constant of desensitization of wild-type channels ($\tau_{wt}$). (**C**) Ratio of sustained/peak currents at time $3x\tau_{wt}$ for various Q276 mutants. Columns in blue or in red represent amino acids smaller or larger than glutamine (Q). Error bars are ± SD, $n$ = 8 to 10 cells. The filled circles superimposed on the bars indicate the volume of the corresponding amino acid in Å[3] as indicated in the y-right axis: in black is Q, in green smaller and in orange larger than Q.

DOI: https://doi.org/10.7554/eLife.45851.002

The following figure supplement is available for figure 1:

**Figure supplement 1.** Representative examples of normalized current traces of various Q276 mutants and the control Q278G.

DOI: https://doi.org/10.7554/eLife.45851.003

open or to brief sojourns to the close state. The scheme of *Figure 2—figure supplement 1* is used to describe kinetics of inactivating channels with strong pre-open inactivation such as Kv4.2 channels (*Bähring and Covarrubias, 2011*), but it could also be applied to describe qualitatively the time course of hASIC1a. Thick arrows indicate a dominant effect of the $k_{ODo}$ rate constant that drives open channels to desensitize completely with a macroscopic time constant of ~1.8 s. $k_{ODo}$ of Q276 mutants must be significantly reduced to explain the long openings. For any reaction loop the clockwise product of rates around the loop must be equal to the counterclockwise product that is other rate constant(s) must also change. Such rate constant is likely $k_{CDc}$. As we will show in the next section, Q276 mutations also markedly decrease SSD that is controlled by $k_{CDc}$. Decreased values of those rate constants contribute to the generation of sustained currents as mutant channels that reach the desensitized states $D_{oH+}$ or $D_{cH+}$ can return to the open state directly or by transitioning

**Table 1.** Summary of calculated values of apparent pH$_{50A}$ of activation, pH$_{50SSD}$ steady-state desensitization and corresponding Hill coefficients: $n$ Overlap of activation and desensitization currents in mutants with decreased desensitization prevented to determine pH$_{50SSD}$ values, they are reported as <pH 6.0 or ND.

Numbers are the mean ± SD of five to eight independent measurements.

|  | pH50a | N | pH50ssd | N |
|---|---|---|---|---|
| WT | 6.69 ± 0.21 | 3.5 | 7.09 ± 0.05 | 10 |
| Q278G | 6.68 ± 0.18 | 3.4 | 7.07 ± 0.02 | 14 |

**Q276 Mutants**

|  | pH50a | n | pH50ssd | n |
|---|---|---|---|---|
| Q276G | 6.67 ± 0.15 | 3.5 | <6.00 | ND |
| Q276A | 6.71 ± 0.12 | 4.2 | <6.00 | ND |
| Q276D | 6.68 ± 0.31 | 4.7 | <6.00 | ND |
| Q276C | 6.61 ± 0.21 | 3.9 | <6.60 | ND |
| Q276E | 6.66 ± 0.18 | 4.2 | 6.80 ± 0.11 | 8 |
| Q276N | 6.64 ± 0.23 | 3.4 | <6.60 | ND |
| Q276L | 6.71 ± 0.13 | 3.8 | 6.6 ± 0.20 | 3 |
| Q276K | 6.55 ± 0.32 | 3.7 | <6.60 | ND |
| Q276R | 6.62 ± 0.23 | 4.1 | <6.30 | ND |
| Q276Y | 6.75 ± 0.11 | 3.8 | <6.20 | ND |
| Q276W | 6.72 ± 0.09 | 3.4 | <6.00 | ND |

**L415 Mutants**

|  | pH50a | n | pH50ssd | n |
|---|---|---|---|---|
| L415G | 4.88 ± 0.11 | 1.3 | <6.00 | ND |
| L415A | 4.50 ± 0.20 | 2 | <6.00 | ND |
| L415D | 5.00 ± 0.11 | 1 | <6.00 | ND |
| L415C | 6.00 ± 0.10 | 2.1 | 7.02 ± 0.05 | 18 |
| L415V | 6.70 ± 0.11 | 6 | 7.02 ± 0.07 | 15 |
| L415E | 6.60 ± 0.10 | 3.1 | 6.90 ± 0.05 | 8.9 |
| L415T | 6.68 ± 0.18 | 2.3 | 6.95 ± 0.04 | 15 |
| L415Q | 6.61 ± 0.11 | 2.6 | 6.82 ± 0.03 | 4.8 |
| L415M | 6.70 ± 0.14 | 6.3 | <6.00 | ND |
| L415R | 6.46 ± 0.21 | 2.3 | <6.00 | ND |
| L415F | 4.41 ± 0.12 | 3.7 | <6.00 | ND |
| L415Y | 4.69 ± 0.22 | 1.5 | <6.00 | ND |

**N416 Mutants**

|  | pH50a | n | pH50ssd | n |
|---|---|---|---|---|
| N416G | 6.50 ± 0.02 | 3.6 | 7.04 ± 0.03 | 7 |
| N416A | 6.50 ± 0.06 | 8 | 6.80 ± 0.01 | 6 |
| N416C | 6.68 ± 0.05 | 5.3 | 7.03 ± 0.05 | 5.6 |
| N416S | 6.58 ± 0.04 | 3.5 | 7.08 ± 0.01 | 7 |
| N416Q | 6.79 ± 0.02 | 7.8 | 6.71 ± 0.06 | 5.6 |
| N416L | 6.85 ± 0.03 | 5.2 | 6.96 ± 0.01 | 4 |
| N416K | 6.60 ± 0.05 | 8 | >6.00 | ND |

*Table 1 continued on next page*

*Table 1 continued*

|  | pH50a | N | pH50ssd | N |
|---|---|---|---|---|
| N416R | 6.22 ± 0.04 | 5 | >6.00 | ND |
| N416Y | 4.85 ± 0.03 | 1.8 | >6.00 | ND |

DOI: https://doi.org/10.7554/eLife.45851.004

first to the closed state (C$_{H+}$) enabling re-openings as long as protons are present. This was indeed observed at the single channel level as indicated above.

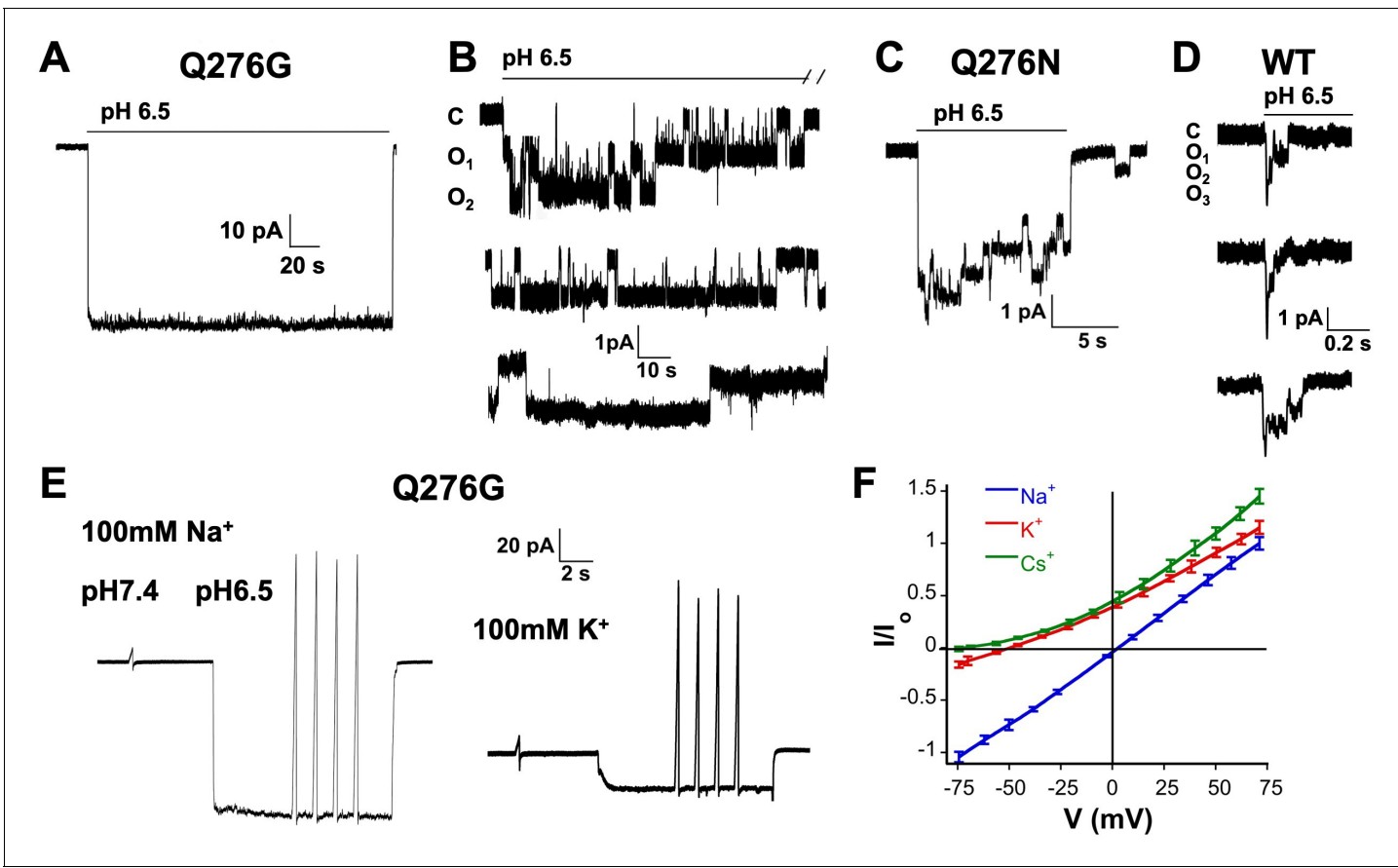

**Figure 2.** Selectivity and kinetics of Q276G and Q276N mutants. (**A**) Representative example of an out-side out patch containing approximately forty Q276G channels. Currents were induced by a rapid change of external solution from pH 7.4 to 6.5 for 80 s and retuned to pH 7.4. Symmetrical 100 mM NaCl. Holding potential −60 mV. (**B**) Example of a patch containing two Q276G channels activated by pH 6.5 continuously for 2.4 min. Some openings last 50 s. C, O$_1$ and O$_2$ indicate the zero current level (closed) and one or two open channels. (**C**) A representative example of out-side out patch containing approximately five Q276N channels activated by pH 6.5. (**D**) Representative examples of unitary currents from wild-type hASIC1a channels elicited by three consecutive sweeps of pH 6.5. An expanded time scale 0.2 s is required to discern opening events. (**E**) Ion selectivity was calculated by changes in reversal potential according to the indicated protocol. Out-side patches expressing Q276G channels were perfused with external solutions containing 100 mM of a selected cation and 100 mM Na$^+$ in the patch pipette. A voltage ramp from −75 to 75 mV of 250 ms duration was applied at pH 7.4. After activation with a solution containing the same external cation but buffered at pH 6.5, ramps were repeated four times for each ion. The currents obtained at pH 6.5 and 7.4 were subtracted to obtained ASIC1a specific currents and to correct for leaks. Only patches with ramps successfully measured with Na$^+$, K$^+$, and Cs$^+$ were used in the calculations. (**F**) I-V relations of Q276G in the presence of 100 mM external Na$^+$, K$^+$ or Cs$^+$. Each line represents the average of three independent patches.

DOI: https://doi.org/10.7554/eLife.45851.005

The following figure supplement is available for figure 2:

**Figure supplement 1.** A kinetic scheme describing gating of channels that exhibit strong desensitization from the pre-open and open states.

DOI: https://doi.org/10.7554/eLife.45851.006

To confirm that the sustained currents of mutant Q276G channels correspond to the natural opening state of hASIC1a and do not arise from alternative pore conformations (*Springauf et al., 2011*), we examined ion selectivity by calculating permeability ratios according to changes in reversal potential in outside-out patches with 100 mM $Na^+$ in the pipette and 100 mM of $Na^+$, $K^+$ or $Cs^+$ in the outside solution (*Figure 2E–F*). The relative permeability ratios were $P_K/P_{Na} = 0.14 \pm 0.01$ and $P_{Cs}/P_{Na} = 0.05 \pm 0.007$. The single channel amplitude measured at $-60$ mV in the presence of symmetric 100 mM $Na^+$ was $1.3 \pm 0.2$ pA that is identical to others and our previous measurements of wild-type channels (*Yang and Palmer, 2014*; *Zhang and Canessa, 2002*). Therefore, Q276 mutants have the same ion permeability ratios and single channel conductance as those of wild type indicating that substitutions of Q276 markedly change channel kinetics without altering permeation properties of the ion pore.

## Q276 mutations confer resistant to proton mediated steady-state desensitization, removal of $Ca^{2+}$ and modulation by polyamines and PcTx1

In addition to slowing or almost abolishing desensitization from the open state, Q276 substitutions also diminished SSD; the apparent $pH_{50SSD}$ of mutants were shifted to more acidic pH and exhibited decreased slope of the curve, whereas wild-type hASIC1a and the control Q278G completely desensitized at preconditioning pH 6.9 (*Figure 3A*; *Table 1*). The most extreme mutant Q276G was entirely insensitive to low preconditioning pH. Shifts of $pH_{50SSD}$ to lower pH were stronger with small and large side chains than with intermediate size substitutions. In contrast, apparent affinities for proton activation, $pH_{50A}$, were similar to wild-type channels (*Figure 3B*; *Table 1*).

$Ca^{2+}$ in the preconditioning solution is known to prevent SSD (*Babini et al., 2002*) because $Ca^{2+}$ ions compete with protons for sites –not yet determined- located in the extracellular domain. We found that $Ca^{2+}$-free solutions did not desensitize Q276G but the absence of $Ca^{2+}$ in the activation solution changed the apparent $pH_{50A}$ from 6.8 to 7.0 (*Figure 3C–D*), indicating that SSD of the mutant is $Ca^{2+}$-independent but modulation of activation remains $Ca^{2+}$ dependent as in wild-type channels (*Paukert et al., 2004*).

Apart from $Ca^{2+}$, other agents can modulate SSD, for instance the tarantula toxin PcTx1 increases sensitivity of protons displacing the $pH_{50SSD}$ to more alkaline values (*Chen et al., 2005*) and the polyamine spermine displaces $pH_{50SSD}$ to more acid pH (*Babini et al., 2002*; *Duan et al., 2011*). In Q276G channels, 5 nM PcTx1 applied at pH 7.4, 7.5 and 7.6, did not inhibit Q276G currents (*Figure 3—figure supplement 1A*), and 0.25 mM spermine did not change SSD but it decreased the $pH_{50a}$ from $6.75 \pm 0.09$ to $6.63 \pm 0.08$ (*Figure 3—figure supplement 1B–C*). Thus far, all the results indicate that side chains different from glutamine in position 276 destabilize or even abolish both high-pH desensitization and SSD without affecting activation, suggesting that Q276 plays a selective role in controlling the desensitization process of hASIC1a.

## The role of Q276 in gating is common to other functional ASIC isoforms

To explore whether the function of Q276 extends to other ASIC isoforms, we mutated the equivalent position to generate rat ASIC2a-Q275G and rat ASIC3-Q269G. Wild-type rat ASIC2a was activated with pH 4.5 because its sensitivity to protons is much lower than that of ASIC1a, and SSD was induced by decreasing the preconditioning pH as indicated in *Figure 4A–B*. Introduction of the equivalent mutation Q275G produced channels with sustain currents that were insensitive to low preconditioning pH. Similar behavior was observed in rASIC3-Q269G (*Figure 4C–D*). Thus, glutamine in β9 has a conserved functional role in the desensitization process of all three functional mammalian proton-sensitive ASIC isoforms.

## Q279 functionally and physically interacts with N416

Crystal structures of cASIC1 show little state-dependent movement of Q276 side chain and β9 strand in the palm domain of cASIC1 implying that the functional effects of this residue most likely arise from interactions with moving neighboring residues. The nearest one in the closed (*Yoder et al., 2018*) (PDB: 5WKV) and open (*Baconguis et al., 2014*) (PDB: 4NTW) conformations is N416, and in the desensitized state is L415 (PDB: 4NYK). To examine whether there is a direct interaction between Q276 and N416 we introduced cysteine residues in the two positions and looked for

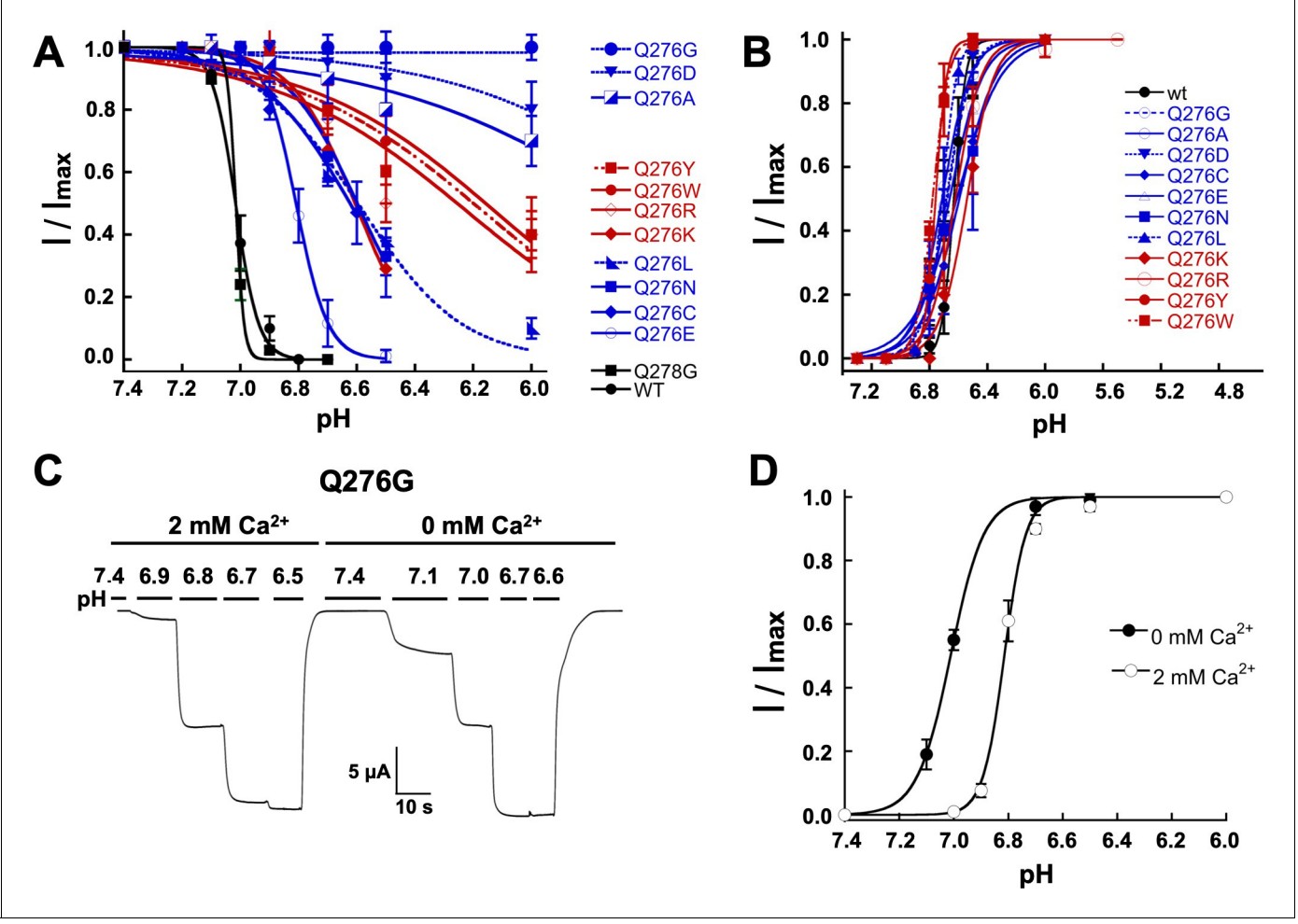

**Figure 3.** Mutations of Q276 diminish SSD. (A) Values of apparent $pH_{SSD}$ and (B) apparent $pH_{50A}$ of substitutions in position 276. Curves represent fits of data points ($n$ = 5–8 independent measurements) to the Hill equation. (C) Current traces of Q276G activated sequentially with solutions of the indicated pH (bars above the trace) in the presence of 2 mM $Ca^{2+}$ and nominal 0 mM $Ca^{2+}$. Pre-conditioning protons do not induce SSD in any of the two conditions. (D) The absence of $Ca^{2+}$ in the activating solution increases the apparent affinity for protons from $pH_{50A}$6.8 to 7.0.

DOI: https://doi.org/10.7554/eLife.45851.007

The following figure supplement is available for figure 3:

**Figure supplement 1.** Q276G channels are insensitive to agents that modulate SSD.

DOI: https://doi.org/10.7554/eLife.45851.008

evidence of spontaneous formation of a disulfide bond. The single mutant Q276C exhibited biphasic currents with peak and sustained components whereas N416C completely desensitized (*Figure 5A*). The double mutant showed mainly sustained currents without SSD (*Figure 5B–C*). After exposure to a reducing agent (5 mM DTT, pH 8.0), the magnitude of the desensitizing current increased and the ratio of peak to sustained component also increased (*Figure 5D*). A possible interpretation of this result would be the presence of a disulfide bond between C276 and C416 that keeps channels in a non-desensitizing conformation but reverts to desensitizing mode after the residues are unrestrained by the reducing agent.

Another means of restricting movement of N416 would be to replace it by bulkier side chains. Substitutions by Q, K, R, and Y exhibited slowly and incompletely desensitizing currents (*Figure 5E–F*) and $pH_{50SSD}$ values shifted to more acid pH (*Figure 5G*; *Table 1*). In the N416K mutant SSD was almost abolished and single channel kinetics showed marked lengthening of opening events -we recorded uninterrupted opening events of 10 to 30 s duration- similar to Q276G (*Figure 5—figure supplement 1B*). Taken together, the results from these experiments suggest that Q276 and N416

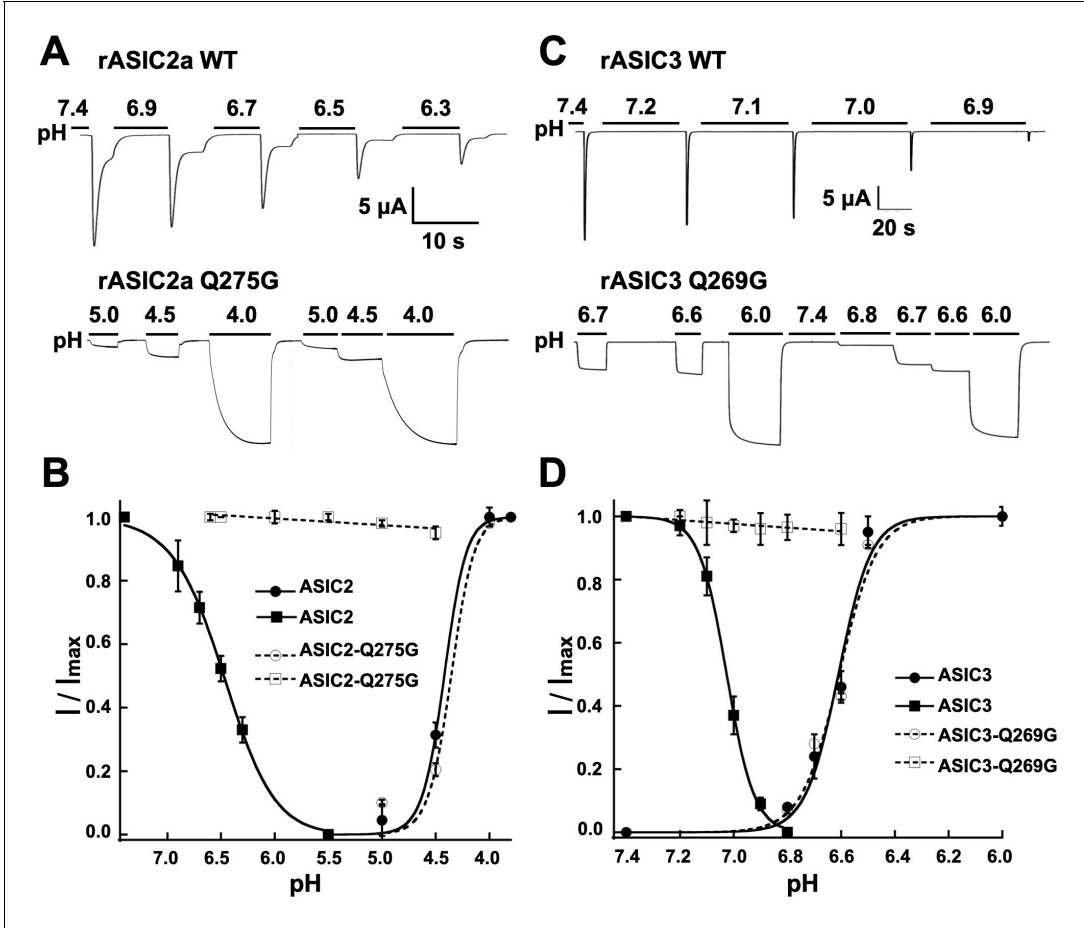

**Figure 4.** The equivalent residue Q276 in hASIC1a controls desensitization in rat ASIC2 and ASIC3. (A) Wild-type rASIC2a currents activated by pH 4.0 undergo progressive desensitization as the preconditioning pH changes from 7.4 to 6.3, whereas mutant Q275G exhibits non-desensitizing currents that are insensitive to low preconditioning pH. (B) pH dependence of activation and SSD of wild-type ASIC2a and ASIC2a-Q275G. Error bars are ± SD, n = 5 cells. (C) and (D) similar experiments conducted with rat ASIC3 and ASIC3-Q269G except for the currents were activated with solution of pH 6.0.
DOI: https://doi.org/10.7554/eLife.45851.009

## A third element in the desensitization mechanism is L415

The position occupied by N416 in the closed and open states is replaced by L415 in the desensitized state as these two residues rotate 180° in opposite directions. Substitutions of L415 also induced sustained currents and shifted pH$_{SSD}$ values to lower pH. The extent of the functional effects is shown in *Figure 5H–J* using the same measurements and association to volume of side chain: residues with small (G, A) and large side chains (R, F, Y) produced the largest changes. Some of these mutants also impaired activation that is current magnitude was small (<1 µA/oocyte) and the pH$_{50A}$ value shifted to more acid pH (*Figure 5—figure supplement 1A–C*), which could be explained by the known role of the linker in transducing conformational changes from the ECD to the pore (*Yoder et al., 2018*). At the single channel level, the sustained currents of L415R exhibited very long openings similar to the kinetics observed in Q276G and N416K (*Figure 5—figure supplement 1D*). As substitutions of Q276, L415 and N416 generate long openings, the transition from open to desensitized state as well as the transitions from pre-open closed to desensitized states must be reduced compared to wild-type channels. Together the findings indicate that these positions produce similar functional effects and size of side chains plays a critical role in the desensitization process suggesting that they may participate in the same mechanism.

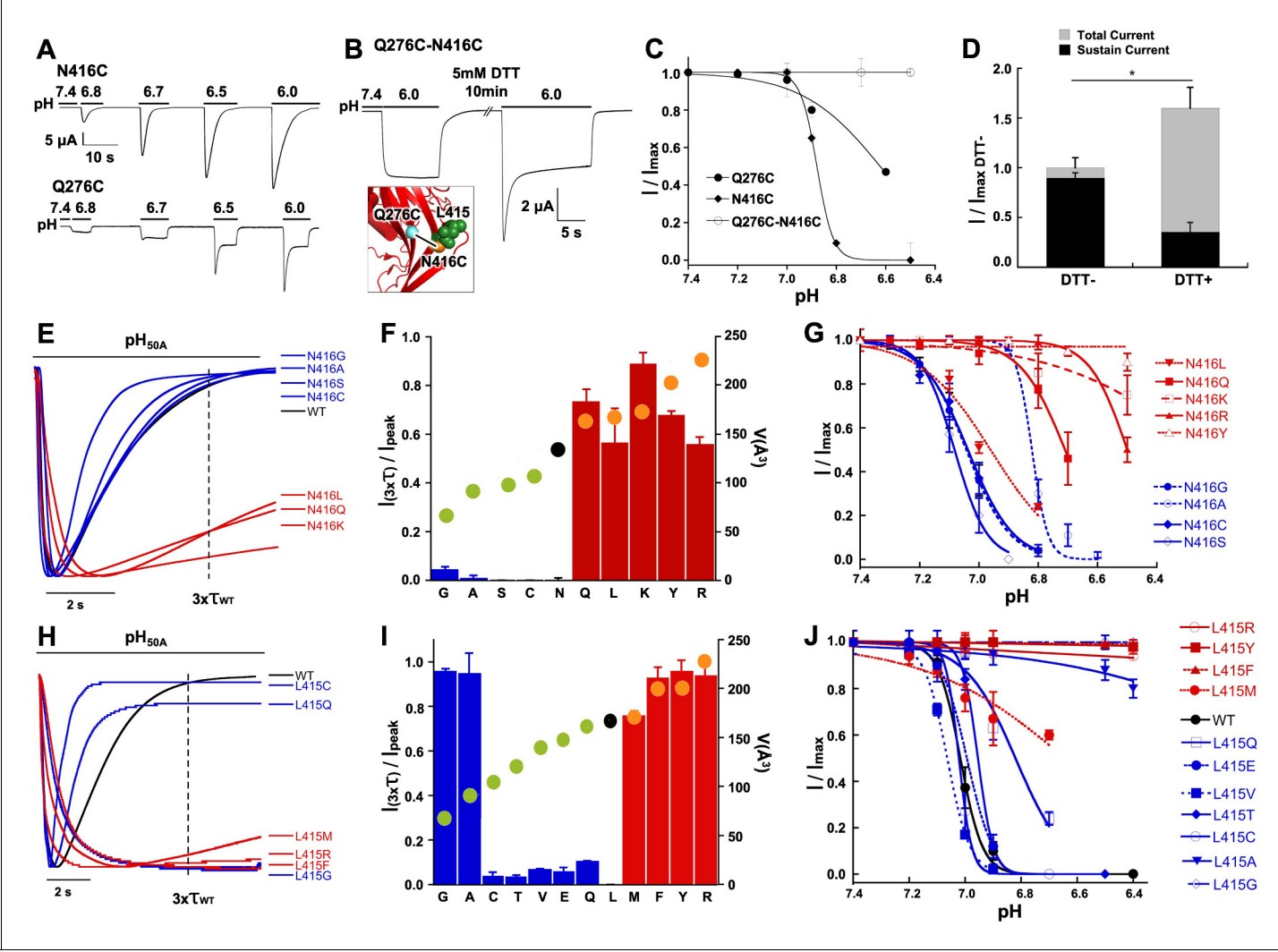

**Figure 5.** Functional and physical interactions of Q276 and N416 residues. (**A**) Current traces of single cysteine mutants N416C and Q276C activated by increasing concentrations of protons. N416C channels completely desensitize, whereas Q276C exhibit a fraction of sustained current. (**B**) The double mutant Q276C-N416C exhibits only sustained currents. After treatment with DTT a desensitizing current appeared. Inset shows predicted locations of the two cysteines in the closed conformation and a putative disulfide bond as a line linking the residues. (**C**) Comparison of the pH dependence of SSD of single and double mutants shows the double mutant is insensitive to SSD. (**D**) Columns represent the fractions of sustained and peak currents before and after DTT treatment. Currents were normalized to the values prior to DTT; n = 15 cells. Asterisks indicate p values < 0.005. (**E**) Normalized currents of substitutions of N416 superimposed on the wild type (black trace). (**F**) Ratio of sustained/peak currents at time $3x\tau_{wt}$ of N416 mutants ordered from low to large side chain volume indicated by the filled circles on the bars. Error bars are SD, n = 8 to 10 cells. (**G**) pH dependence of SSD of N416 mutants. Lines are the fit of data points (n = 5–8 independent measurements) to the Hill equation ± SD. (**H**) Normalized currents of substitutions of L415 superimposed on the wild type (black trace). (**I**) Ratio of sustained/peak currents at time $3x\tau_{wt}$ of L415 mutants ordered from low to large side chain indicated by the filled circles on the bars. Error bars are SD, n = 8 to 10 cells. (**J**) pH dependence of SSD of L415 mutants. E data point is the mean ± SD of 5 to 8 independent cells.

DOI: https://doi.org/10.7554/eLife.45851.010

The following figure supplement is available for figure 5:

**Figure supplement 1.** Mutations in the b11-b12 linker decrease sensitivity to protons and prolonged duration of openings.
DOI: https://doi.org/10.7554/eLife.45851.011

## SSD arises from asynchronous proton-induced desensitization of a single subunit in the hASIC1a trimer

As already indicated, a distinct feature of hASIC1a is complete desensitization at low proton concentrations, pH 7.1 to 7.0. In that pH range channels do not open upon proton binding but instead

enter the desensitized state (transitions from $C_{H+}$ to $D_{CH}$ in *Figure 2—figure supplement 1*). Here, we asked how many subunits in the hASIC1a channel must undergo such transition in order to induce SSD. To explore this question we compared the induction of SSD in channels formed by three wild-type hASIC1a subunits to channels containing one, two or three non-desensitizing Q276G subunits. The desired stoichiometry was kept constant in concatemers composed of WT-Q276G-WT or WT-Q276G-Q276G subunits (*Figure 6—figure supplement 1*). Trimers with a single mutated subunit exhibited only peak currents that completely desensitized at low pH and exhibited SSD similar to that of wild-type channels. Trimers containing two mutated subunits exhibited a large fraction of completely desensitizing peak current and a small component (0.05 ± 0.03) of sustained current (*Figure 6A*). The apparent $pH_{50SSD}$ values of WT-Q276G-WT and WT-Q276G-Q276G were similar to that of wild type (*Figure 6B*). Therefore, a conformational change of a single subunit in ASIC, induced by low proton concentrations, in the pH range of 7.1 to 7.0, is sufficient to produce SSD.

## Discussion

Desensitization shapes the time course of ASIC currents and attenuates depolarization of neurons mediated by prolonged increases in proton concentrations. The mechanisms to achieve desensitization vary and are distinct to each class of channel, either voltage- or ligand-gated, but all are designed to shut the pore by closing either the opening gate or a secondary gate.

Significant insight in the molecular mechanism mediating proton gating of ASICs has been obtained from the crystal structures of cASIC1 in close, open and low-pH desensitized conformations. However, a mechanism describing desensitization, both SSD and low-pH desensitization, is not yet fully resolved. Here, we found that residue Q276 located in the rigid β9 strand -that does not undergo movements between conformations- has a profound effect on the desensitization process of ASIC1a, ASIC2a and ASIC3s, the three functional mammalian isoforms. The size of the side chain in this position relates to the magnitude of the functional effects: if very large or very small desensitization is impaired and in the Q276G mutant it is mostly abolished. Underlying this effect is a marked lengthening of the opening events consistent with Q276 determining the transition rate from the open to the desensitized state. Noticeable, other channel properties such as proton-mediated activation, ion selectivity and conductance were not affected indicating that Q276 functions

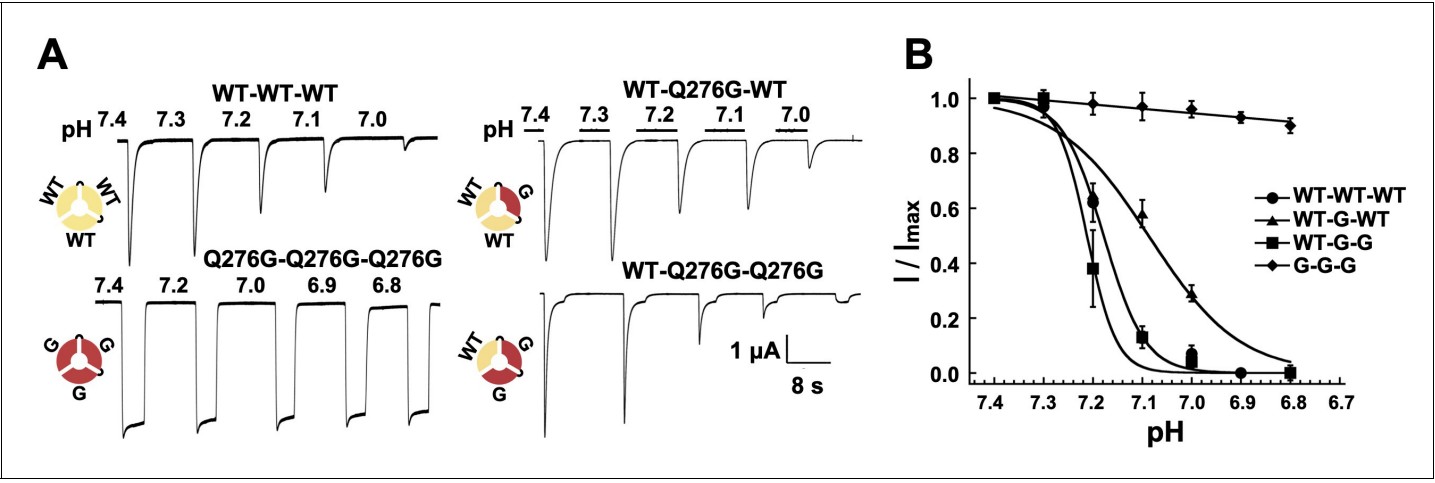

**Figure 6.** One WT subunit in ASIC1a trimer is sufficient to induce SSD. (A) Current traces of four trimers containing three wild-type subunits (WT-WT-WT), one (WT-Q276G-WT) or two (WT-Q276G-Q276G) mutant subunits elicited by pH 6.5 exhibit SSD but not channels with three (Q276G-Q276G-Q276G). WT-Q276G-Q276G trimers exhibit a small component of current resistant to desensitization. (B) pH response to SSD of the four trimers shown in (A).

DOI: https://doi.org/10.7554/eLife.45851.012

The following figure supplement is available for figure 6:

**Figure supplement 1.** Schematic and expression of trimers containing wild type or mutant monomers (Q276G).

DOI: https://doi.org/10.7554/eLife.45851.013

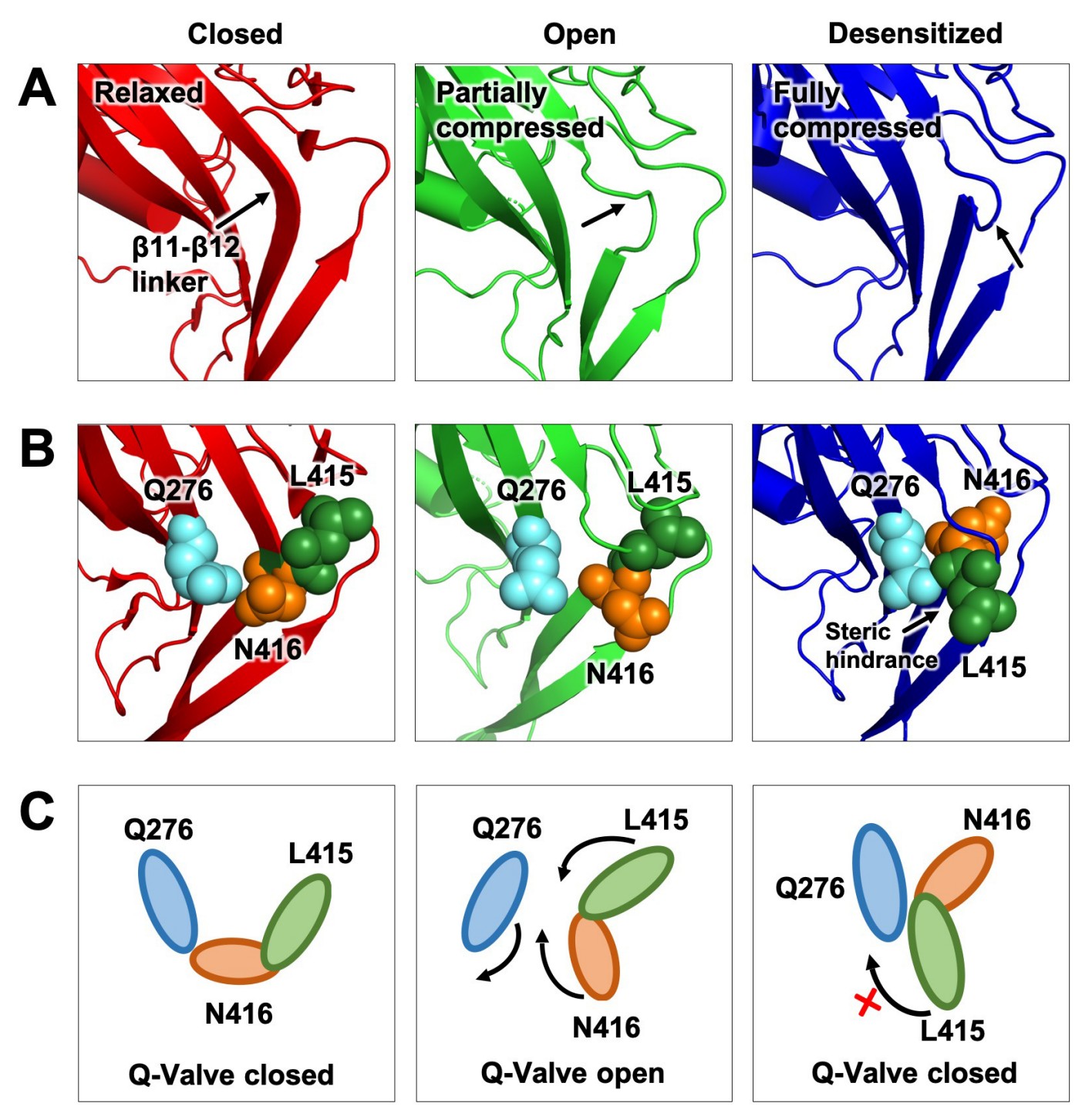

**Figure 7.** Proposed Q-valve mechanism to control of hASIC1a desensitization. (**A**) Ribbon representation of the area containing the Q-valve: β9 and β11- β12 linker in the ECD of cASIC1 in Closed, Open and Desensitized conformations. The linker is compressed upon proton activation inducing 180° rotation of L415 and N416. Slight upward movement of β12 further increases the compression in the desensitized channel. (**B**) Side chains of Q276, L415 and N416 are shown as colored spheres. (**C**) Cartoon depicting proposed mechanism of the Q-valve: in the Closed state Q276 and L415 close the valve. In the Open state, the valve opens as protons induce a conformational change of the upper ECD that compresses the upper vestibule formed by β strands of the palm and the β11- β12 linker. Compression of the linker promotes an 180° rotation of N416 and L415 in opposite directions and facing Q276: trajectory indicated by the arrows. Red cross shows the valve closed and how Q276 could prevent backward rotation of L415, which in turn would relax the linker preventing full desensitization. Predicted intermediate states and direction of rotation are derived from functional analysis of substitutions in the three key residues making the valve.

*Figure 7 continued on next page*

*Figure 7 continued*

DOI: https://doi.org/10.7554/eLife.45851.014
The following video is available for figure 7:
**Figure 7—video 1.** From analysis of the functional data of mutant channels and cASIC1 structures it emerges a possible mechanism for desensitization wherein the central elements are three residues: Q276 located in the middle of β9 strand and L415 and N416 in the β11- β12 linker.
DOI: https://doi.org/10.7554/eLife.45851.015

specifically in the desensitization process. We also show that substitutions of the two residues L415 and N416 located in close proximity to Q276 in the β11- β12 linker also disrupt desensitization by lengthening the opening events and that size of the side chains is also an important factor in determining the magnitude of the effects: small and large side chains were more effective in disrupting desensitization, in a pattern similar to substitutions of Q276. Contrary to Q276, some mutations in the linker also modified proton-gating namely shifted the pH$_{50A}$ to more acidic range (*Figure 5—figure supplement 1*) likely owing to the β11- β12 linker participation in transducing conformational changes from the upper ECD to the pore gate during proton-mediated activation.

To explain the prominent role of Q276 in desensitization and similar functional effects of substitutions in the linker we propose a molecular mechanism for desensitization wherein the side chain of Q276 operates as a valve that allows or restricts rotation of L415 and N416. In the proposed mechanism, the pathway of movement of L415 and N416 faces Q276 that is it is inward rather than outwards as was originally suggested (*Yoder et al., 2018*). This configuration enables transient functional interactions of the three key residues and makes the mechanism highly reliant on their respective sizes. A competent valve –one that ensures complete desensitization- requires concerted work of the three conserved residues; changing any of them introduces steric hindrance that leads to a leaky valve manifested as incomplete or absent desensitization. This is illustrated in *Figure 7* and the movie of *Figure 7—video 1*. An inward pathway of the rotation also provides a means of locking the valve to prevent back swings during desensitization. Such a block would not be present if the trajectory is in the outward direction.

This mechanism applies not only to desensitization from the open state but also from pre-open closed states that exist at pH 7.2 to 7.0 (*Figure 1—figure supplement 1*). In this pH range, protons bind in average to only one subunit of the trimer or at most two subunits. The protonated subunit(s) may adopt the open conformation or another as yet unknown conformation; however, steric hindrance imposed by non-protonated neighboring subunits may prevent opening of the gate. Thus, at low proton concentrations, there is predominantly asynchronous protonation of subunits leading to SSD rather than channel openings (left loop *Figure 2—figure supplement 1*). In contrast, at high proton concentrations (pH 6.5–6.0), the three subunits bind protons simultaneously enabling concerted conformational changes of the subunits that lead to channel opening. Results from trimers with fixed stoichiometry of wild type and Q276G subunits support this notion, as the presence of a single wild type subunit in the hASIC1a channel was sufficient to induce complete SSD.

Overall our results are consistent with previous reports of mutations and modifications of cysteine substitutions in the β1-β2 (*Coric et al., 2003*) and β11-β12 linkers (*Springauf et al., 2011*), in the β1 strand forming the lower palm (*Cushman et al., 2007*; *Roy et al., 2013*), TM1 (*Li et al., 2010a*) and even the thumb domain that makes contact with the upper part of TM1 (*Krauson and Carattino, 2016*). Recent structural and functional data have informed that those regions are the ones that undergo the largest conformational changes, and all are located between the Q-valve and the pore, consistent with transducing conformations to the gate. Hence, experimentally introduced mutations or evolution guided species-specific substitutions in any of those domains can alter desensitization and even enable sustained currents (as indicated in the kinetic schemes of *Figure 2—figure supplement 1*). This explains why not all channels that have the tried Q276-L415-N416 exhibit identical kinetics of desensitization.

This study provides experimental support for the original suggestion that the β11- β12 linker is essential for ASIC1 desensitization and that it disengages the upper and lower ECD; however, we found that it does not work alone but needs residue Q276 to keep the linker compressed during desensitization.

In summary, we combined the insight drawn from the structures together with functional analysis of key residues to propose a dedicated and parsimonious mechanism to control desensitization of

ASICs. Further proof of this mechanism could be obtained by structural analysis of the mutant Q276G channel that remains in the proton-induced open conformation for long times. Such analysis could reveal the linker caught in intermediate states as predicted by Q-valve mechanism.

# Materials and methods

## Key resources table

| Reagent type (species) or resource | Designation | Source or reference | Identifiers | Additional information |
|---|---|---|---|---|
| Antibody | Anti-V5 HRP Mouse monoclonal | ThermoFisher Scientific | 46–0708 | Dilution 1:2000 |
| Antibody | Anti FLAG M2 HRP Mouse monoclonal | Sigma | A8592-.2MG | Dilution 1:2000 |
| Antibody | HA-probe (F-7) Mouse monoclonal | Santa Cruz Biotechnology | sc-7392 | Dilution 1:2000 |
| Antibody | Mouse IgG Secondary Antibody HRP conjugate | ThermoFisher Scientific | A16072 | Dilution 1:10000 |
| Recombinant DNA reagent (DNA plasmid) | pcDNA3.1-hASIC1a | PMID: 23048040 | | Cloned from HEK293T |
| Recombinant DNA reagent (DNA plasmid) | pcDNA3.1-ratASIC2a | PMID: 11382806 | | Cloned from rat dorsal root ganglia |
| Recombinant DNA reagent (DNA plasmid) | pcDNA3.1-ratASIC3 | PMID: 11382806 | | |
| Recombinant DNA reagent (DNA plasmid) | pcDNA3.1- h ASIC1aHA-hASIC1aV5-hASIC1aFl | This paper | | Expression plasmid wt hASICa trimer |
| Recombinant DNA reagent (DNA plasmid) | pcDNA3.1-hASIC1aQ276G HA-hASIC1aQ276GV5-hASIC1a Q276GFl | This paper | | Expression plasmid hASIC1aQ276G trimer |
| Recombinant DNA reagent (DNA plasmid) | pcDNA3.1-h ASIC1aHA-hASICQ276GV5-hASIC1aFl | This paper | | Expression plasmid one mutant subunit Q276G |
| Recombinant DNA reagent (DNA plasmid) | pcDNA3.1-hASIC1aHA-hASICQ276GV5-hASIC1aQ276GFl | This paper | | Expression plasmid two mutant subunits Q276G |
| Peptide, | Psalmotoxin-1 | Alomone labs | STP-200 | five nM |
| Commercial assay or kit | QuickChange Site-Directed Mutagenesis Kit | Agilent | 200523 | Mutagenesis |
| Commercial assay or kit | mMESSAG EmMACHINE T7 | ThermoFisher Scientific | AM1344 | Synthesis of cRNA |
| Commercial assay or kit | Pierce BCA Protein Assay Kit | ThermoFisher Scientific | CN: 23227 | Protein quantification |
| Chemical compound, drug | Spermine | SIGMA-ALDRICH | S3256-1G | 0.25 mM |
| Chemical compound, drug | Tris (2-carboxyethyl) phosphine hydrochloride | SIGMA-ALDRICH | C4706 | 5 mM |
| Software, algorithm | PyMOL | | RRID:SCR_000305 | |
| Software, algorithm | KaleidaGraph | | RRID:SCR_014980 | |

## Site-directed mutagenesis of hASIC1a, rASIC2a and rASIC3

Mutations were introduced in cDNA using QuickChange according to the manufacturer instructions. All cDNAs were sequenced to verify the correct mutations. hASIC1a trimers were constructed by ligating subunits with unique tags in the carboxyterminus of each monomer and restriction sites using PCR. The integrity of the constructs was verified by DNA sequencing and probed in western blots using HA, V5 and Flag antibodies.

### Western blotting

Cells expressing hASIC1 trimers were homogenized with buffer containing (mM): 150 NaCl, 50 Tris-base ph 7.4, 5 EDA, 1% Triton X-100 on ice for 15 min. Lysates were cleared by centrifugation 10 K rpm at 4°C. After protein quantification (Pierce BCA Protein Assay Kit), 30 µg of protein were loaded on SDS-10% polyacrylamide gels. Proteins were transferred to membranes (Immobilon-P CN; IPVH00010, Millipore) by electrophoresis and subsequently processed for blotting with HA (Santa Cruz Biotechnologies), Flag-HRP (SigmaAldrich) or V5-HRP (ThermoFisher Scientific) mouse monoclonal antibodies. After washes, the blot using HA monoclonal received anti-mouse secondary antibody labeled with HRP (ThermoFisher Scientific).

### Isolation and injection of *Xenopus* oocytes

Female *Xenopus laevis* were obtained from the in house breeding facility from professor Tao Qinghua at Life Sciences, Tsinghua University, Beijing. The Association for Assessment and Accreditation of Laboratory Animal Care International (AALAC) has accredited this animal facility. Frog surgery was conducted according to the protocol approved by IACUC of Tsinghua University (protocol number 07749). After treatment of oocytes with collagenase and extensive washes, they were injected with 5 ng of cRNA made with mMESSAGEmMACHINE T7. Oocytes were kept at 18°C and used at room temperature 24–48 hr post-injection.

### Two-electrode voltage clamp

Whole-cell currents were measured using a two-electrode voltage clamp (Oocyte-Clamp OC-725C, Warner Instrument Corp.) with PowerLab 8/35 (ADInstruments) running LabChart Prosoftware. Cells were placed in a fast exchange perfusion chamber with high flow delivered by gravity. Perfusion solutions had the following composition in mM: 100 NaCl, 4 KCl, 2 CaCl$_2$, 5 HEPES, 5 MES, pH was adjusted to desired values with n-methyl-d-glucamine. When indicated, CaCl$_2$ was omitted from the solutions. Pipette resistances were 0.5–1 MΩ when filled with 3M KCl. Oocytes were voltage clamped at −60 mV unless indicated. ASIC currents were activated by changing the external solution from a conditioning pH of 7.4 -or any other indicated value- to a more acidic test pH for 5 s to 30 s until currents completely desensitized –that is reached the zero current level- or the sustained current reached a plateau. The external solution was returned to pH 7.4 or changed to the indicated pre-conditioning pH for 15 to 30 s, short time for non-desensitizing and long for desensitizing mutants. To calculate the pH$_{50a}$, each mutant channel was first activated with solutions in the pH range from 6.9 to 4.5. Lower pH values induced a sustained endogenous current in oocytes. Most channels reached Imax around pH 6.0 to 6.5, peak currents decreased as pH was lowered further. Thus, we used a pH range from no activation to the one that induced maximal current for the mutant tested. Channels with pH$_{50a}$ markedly shifted to the right, the range was extended to pH 4.5. Psalmotoxin-1 was from Alomone Labs, Tris(2-carboxyethyl)phosphine (TCEP) and other chemicals were from Sigma-Aldrich.

### Patch clamp

hASIC1a currents were recorded in excised patches from oocytes in the outside-out configuration using a HEKA patch clamp EPC10 amplifier and PATCHMASTER acquisition software v2 × 90.2 (HEKA Electronic). Pipette solution contained in mM: 100 KCl, 20 HEPES pH 7.4. Bath solution contained 100 NaCl, 4 KCl, 2 CaCl$_2$, 20 HEPES/MES adjusted to pH 7.4 or 6.5 with N-methyl-D-glucamine. Membrane potential was held at −60 mV. Patches were perfused with a solution of pH 7.4 to establish the baseline current, followed by activation with a solution of pH 6.5 using a fast-exchange perfusion system (SF-77B perfusion-step, Warner Instruments). Experiments were conducted at

room temperature. For determination of ion permeabilities, 100 mM NaCl was replaced with 100 mM KCl or CsCl. $CaCl_2$ was omitted from the solutions in these experiments. For measurements of I-V relationships, the holding potential was changed from $-75$ mV to 75 mV in ramps of 250 ms duration that were repeated three consecutive times.

## Data analysis

I-V relationships were calculated by subtracting background currents measured at pH 7.4 from those measured during activation of channels at pH 6.0 using the same cation in the perfusion of out-side out patches. Permeability ratios were calculated from the shift of the reversal potential ($\Delta V_{rev}$) of the I-V relationship when $Na^+$ in the perfusion solution was substituted by another ion X according to: $P_x/P_{Na} = exp(F \bullet \Delta V_{rev}/RT)$. Slope conductance was measured from I-V plots over the voltage rage of $-60$ to 0 mV.

Concentration response curves were fit to the Hill-Function: $I = 1/(1+(EC_{50}/[H^+]^n)$, where $EC_{50}$ is the pH at which the half-maximal activation/desensitization of the maximal current was achieved, and $n$ is the Hill coefficient. Time constants of desensitization were determined by fitting the decay of the current traces to a single exponential. The results are reported as the means ± SD. They represent the mean of five to eight individual measurements in different oocytes. Statistical significant differences between groups were determined with Student's unpaired $t$ test.

### Structure alignments

Crystal structures of cASIC1a in resting (PDB: 5WKV), open (PDB: 4NTW) and desensitized (PDB: 4NYK) were downloaded from Protein Data Bank. Single-subunit superpositions were done with PyMOL program protein alignment.

## Acknowledgements

The work was supported by a 1000 Talent Program Grant to CC from the People's Republic of China and by the National Institute of Health 5R21MH10746402.

## Additional information

### Funding

| Funder | Grant reference number | Author |
|---|---|---|
| National Institute of Mental Health | 5R21MH10746402 | Cecilia M Canessa |
| Thousand Talents Plan | | Cecilia M Canessa |

The funders had no role in study design, data collection and interpretation, or the decision to submit the work for publication.

### Author contributions

Yangyu Wu, Cecilia M Canessa, Conceptualization, Data curation, Formal analysis, Supervision, Funding acquisition, Investigation, Writing—original draft, Project administration, Writing—review and editing; Zhuyuan Chen, Data curation, Formal analysis, Investigation, Writing—original draft

### Author ORCIDs

Cecilia M Canessa (iD) http://orcid.org/0000-0003-0536-2679

### Ethics

Animal experimentation: This study was performed in strict accordance with the recommendations in the Guide for the Care and Use of Laboratory Animals of the People's Republic of China. The Association for Assessment and Accreditation of Laboratory Animal Care International (AALAC) has accredited this animal facility. Frog surgery was conducted according to the protocol approved by

IACUC of Tsinghua University (protocol number 07749) performed under ethyl 3-aminobenzoate methanesulfonate anesthesia, and every effort was made to minimize suffering.

## Decision letter and Author response

Decision letter https://doi.org/10.7554/eLife.45851.024
Author response https://doi.org/10.7554/eLife.45851.025

## Additional files

### Supplementary files

• Transparent reporting form
DOI: https://doi.org/10.7554/eLife.45851.016

### Data availability

All data generated or analysed during this study are included in the manuscript and supporting files.

The following previously published datasets were used:

| Author(s) | Year | Dataset title | Dataset URL | Database and Identifier |
|---|---|---|---|---|
| Yoder N, Gouaux E | 2018 | Structure of an acid sensing ion channel in a resting state with calcium | https://www.rcsb.org/structure/5WKV | Protein Data Bank, 5WKV |
| Baconguis I, Bohlen CJ, Goehring A, Julius D, Gouaux E | 2014 | Structure of acid-sensing ion channel in complex with snake toxin | https://www.rcsb.org/structure/4NTW | Protein Data Bank, 4NTW |
| Baconguis I, Bohlen CJ, Goehring A, Julius D, Gouaux E | 2014 | Structure of a membrane protein | https://www.rcsb.org/structure/4NYK | Protein Data Bank, 4NYK |

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
