## [Decision Letter]

Thank you for submitting your article "A valve-like mechanism controls desensitization of functional mammalian isoforms of acid-sensing ion channels" for consideration by *eLife*. Your article has been reviewed by three peer reviewers, and the evaluation has been overseen by Kenton Swartz as the Reviewing Editor and Richard Aldrich as the Senior Editor. The following individuals involved in review of your submission have agreed to reveal their identity: Stephan A Pless (Reviewer #1); Toshimitsu Kawate (Reviewer #2).

The reviewers have discussed the reviews with one another and the Reviewing Editor has drafted this decision to help you prepare a revised submission.

Summary:

This study investigates the origin and mechanism of desensitization from open and pre-open states (steady-state desensitization, SSD) in Acid Sensing Ion channels (ASICs). The key finding is that some mutations at three conserved residues Q276, L415 and N416 can virtually abolish both types of desensitization, with minimal impact on other parameters, such as pH50, conductance or ion selectivity. The authors also show that at least the role of Q276 is conserved across other mammalian isoforms, suggesting a broadly applicable mechanism. Lastly, they propose that a single WT subunit is sufficient to induce both types of desensitization. Although a recent structural study had already suggested the involvement of at least L415 in desensitization from open states, the present work is novel and of high relevance to the field. Not least by strongly suggesting that both types of desensitization appear to have the same physical origin and/or converge on to the same pathway. In the past, the lack of ASIC mutations that strongly inhibit or even abolish desensitization has been a major hindrance in studying state-dependence in e.g. pharmacological studies or those focusing on functional mechanisms. Therefore, the results of the present study are likely to enable numerous future studies. Undertaking the following essential and minor revisions will considerably strengthen the manuscript.

Essential revisions:

1) The authors propose a relatively detailed mechanism by which they envision desensitization to occur. However, we are not convinced their functional data (even in combination with the available structural data) allows them to draw these detailed conclusions. For example, the authors state "…changing any of them ((G276, L415, N416)) introduces steric hindrance that leads to a leaky valve.." and "…makes the mechanism reliant on their respective sizes."…, yet for example L415G/A are just effective at preventing desensitization as are L415F/Y/R. The conclusions regarding the details of the mechanism should therefore be revised to better reflect the data. Additionally, it is not clear why they coin the term "Q valve" with reference to only the role of Q276, when mutations at positions L415 and N416 can have very similar effects. This should be reconsidered and possibly revised to something like "QLN desensitization gate/hub" or similar. In addition, it would be good for the authors to discuss alternative possible mechanisms.

2) The authors use concatemeric constructs to assess how many WT subunits are necessary to observe full desensitization. But there are two issues with their results: first, they need to show what happens functionally in the Q276G-Q276G-Q276G triple mutant trimer (and the WT-WT-WT trimer). Based on their findings, one would expect Q276G-Q276G-Q276G to resemble that of the regular Q276G mutant, but concatemeric ion channel constructs are notorious for producing unexpected results (https://www.ncbi.nlm.nih.gov/pubmed/12488557; https://www.ncbi.nlm.nih.gov/pubmed/29382698). This is particularly important in light of the second issue, which is that the present functional and biochemical data (in Figure 6 and Figure 6—figure supplement 1) does not exclude the possibility of a significant population of partial concatemers, i.e. individual WT or mutant subunits that result from incomplete translation. Another very valuable control would be to show that a single WT subunit supports desensitization independently of whether it is present in the first, second or third protomer.

3) The current version of the manuscript does not contain any tables for pH50 or Hill coefficient values for activation and SSD. They need to be included, in particular because some of the data shown in a number of figure panels are virtually impossible to distinguish (e.g. Figure 3A/B; Figure 5E/G/H/J; Figure 5—figure supplement 1A/C etc). "Additional representative examples of other mutants are shown in Figure 1—figure supplement 1." Additional examples of the same mutants shown in Figure 1 do not add much. It would be helpful if the authors include representative traces of the ones that are excluded in Figure 1 (e.g. Q276L). There is no figure legend for Figure 5H-J. It is also not clear how many cells were tested in some cases. Please add n values to all the figure legends, including supplemental figures and tables.

4) "Nevertheless, the duration of mutant openings was markedly longer than that of wild type channels…" It is difficult to gauge what "markedly longer" means without a wild type control. A representative wild type trace in Figure 2 would be helpful. The same is also true for Figure 3—figure supplement 1. We understand that the properties of WT ASICs have been reported previously, but WT controls under the same experimental conditions are necessary for evaluating the effects of the Q276G mutation.

5) Figure 3A and others; it is unclear how those SSD experiments were done (e.g. duration of preconditioning, at what pH the reported current was measured, how Imax was obtained, etc). Include a brief description in figure legends and a more thorough methodology in Materials and methods.

6) Generally, the Discussion section should place the work into a wider context. However, in its present form, the discussion lacks detail and should be expanded to include a more in-depth discussion on previous work (https://www.ncbi.nlm.nih.gov/pubmed/12947112; https://www.ncbi.nlm.nih.gov/pubmed/17389250), including comments or speculation on mechanisms of desensitization in non-mammalian ASICs (https://www.ncbi.nlm.nih.gov/pubmed/20064854).

---

## [Author Response]

Essential revisions:1) The authors propose a relatively detailed mechanism by which they envision desensitization to occur. However, we are not convinced their functional data (even in combination with the available structural data) allows them to draw these detailed conclusions. For example, the authors state "…changing any of them ((G276, L415, N416)) introduces steric hindrance that leads to a leaky valve.." and "…makes the mechanism reliant on their respective sizes."…, yet for example L415G/A are just effective at preventing desensitization as are L415F/Y/R. The conclusions regarding the details of the mechanism should therefore be revised to better reflect the data.

The proposed desensitization mechanism is composed of three elements Q276, L415 and N416 that dynamically interact: Q276 is on the rotation trajectory of L415 and N416. Each of these residues rotates in opposite direction phasing Q276. The data show that if the side chain in position 276 is larger or smaller than Q, rotations of the two residues in the β11- β12 linker (L4215, N416) are impaired in various degrees. For instance, if the side chain in 415 is very large (F, Y, W) steric hindrance prevents full rotation diminishing desensitization whereas very small side chains (G, A) enable backward rotation because they cannot be locked by Q276 in the desensitized conformation. This makes the valve ‘leaky’. For position N416, large residues here produce steric hindrance decreasing desensitization whereas small residues can pass without obstruction. Because N416 does not participate in locking the valve, small residues are tolerated in this position i.e. they enable desensitization. The cartoon of Figure 7C and the supplementary video illustrate how the proposed valve mechanism operates and predicts the functional effects of mutations in each of the three elements of the valve.

Additionally, it is not clear why they coin the term "Q valve" with reference to only the role of Q276, when mutations at positions L415 and N416 can have very similar effects. This should be reconsidered and possibly revised to something like "QLN desensitization gate/hub" or similar. In addition, it would be good for the authors to discuss alternative possible mechanisms.

We chose the term valve-like mechanism as a simple and short way to describe how the three essential residues operate together and to highlight the central role of Q276, which is a residue that does not tolerate substitutions i.e. all other amino acids impair in some degree desensitization.

From the English dictionary a ‘valve’ is a device that regulates the flow of gases, liquids or other materials through apertures by opening, closing or obstructing passageways. We thought that the most descriptive and accurate term is a valve because Q276 regulates the movement of L415 and N416 but Q276 is not an element that moves in the proposed mechanism. We cannot argue other ways to describe the relative movements of these residues; we just have not come up with better words to describe it.

2) The authors use concatemeric constructs to assess how many WT subunits are necessary to observe full desensitization. But there are two issues with their results: first, they need to show what happens functionally in the Q276G-Q276G-Q276G triple mutant trimer (and the WT-WT-WT trimer). Based on their findings, one would expect Q276G-Q276G-Q276G to resemble that of the regular Q276G mutant, but concatemeric ion channel constructs are notorious for producing unexpected results (https://www.ncbi.nlm.nih.gov/pubmed/12488557; https://www.ncbi.nlm.nih.gov/pubmed/29382698). This is particularly important in light of the second issue, which is that the present functional and biochemical data (in Figure 6 and Figure 6—figure supplement 1) does not exclude the possibility of a significant population of partial concatemers, i.e. individual WT or mutant subunits that result from incomplete translation. Another very valuable control would be to show that a single WT subunit supports desensitization independently of whether it is present in the first, second or third protomer.

We have added to Figure 6 concatamers made up of three WT or three Q276G mutant subunits. Wt-wt-wt and Q276G-Q276G exhibit currents with properties identical to channels made by injection of single subunits. We have also replaced the previous western blot in Figure 6—figure supplement 1 with a new one that contains the four concatamers blotted with three antibodies (each for one of the monomers in the concatamer): anti-Fl, anti-V5 and anti-HA in order to visualize any abnormal size products that could represent monomers or dimers. As demonstrated in the blots, only bands of molecular size corresponding to a trimer -three hASIC1a in a single protein- were detected.

3) The current version of the manuscript does not contain any tables for pH50 or Hill coefficient values for activation and SSD. They need to be included, in particular because some of the data shown in a number of figure panels are virtually impossible to distinguish (e.g. Figure 3A/B; Figure 5E/G/H/J; Figure 5—figure supplement 1A/C etc). "Additional representative examples of other mutants are shown in Figure 1—figure supplement 1." Additional examples of the same mutants shown in Figure 1 do not add much. It would be helpful if the authors include representative traces of the ones that are excluded in Figure 1 (e.g. Q276L). There is no figure legend for Figure 5H-J. It is also not clear how many cells were tested in some cases. Please add n values to all the figure legends, including supplemental figures and tables.

A table listing all examined mutants with the respective values of apparent pH50a and pH50ssd has been included. For mutants with markedly decreased proton affinity for SSD the values are reported as <pH 6.0 and the n could not be estimated owing to overlap of activation and desensitization currents.

We have added representative traces of mutants Q276A and Q276L in Figure 1—figure supplement 1. Figure legends of Figure 5 H-J have been included. In the Materials and methods section under Data analysis is indicated that each data point represents the average of 5-8 independent cells.

4) "Nevertheless, the duration of mutant openings was markedly longer than that of wild type channels…" It is difficult to gauge what "markedly longer" means without a wild type control. A representative wild type trace in Figure 2 would be helpful. The same is also true for Figure 3—figure supplement 1. We understand that the properties of WT ASICs have been reported previously, but WT controls under the same experimental conditions are necessary for evaluating the effects of the Q276G mutation.

We include three consecutive sweeps of a patch expressing hASIC1a channels activated by pH 6.5 i.e., same conditions and equivalent number of channels – three channels- as the mutant examples shown in Figure 2B-C. Seven opening events can be discerned from the three traces; they have durations ranging from 4 to 230 ms with a mean of 83.5 ± 99 ms and are followed by complete desensitization concordant with previously published results and are in sharp contrast to the duration of mean open times of the mutants. The new data are shown in Figure 2D. Notice that the time scale for WT is 0.2 s whereas for Q276G and Q276N is 10s and 5s, respectively.

5) Figure 3A and others; it is unclear how those SSD experiments were done (e.g. duration of preconditioning, at what pH the reported current was measured, how Imax was obtained, etc). Include a brief description in figure legends and a more thorough methodology in Materials and methods.

The following paragraph has been added to the Materials and methods section under Two Electrode Voltage Clamp. It describes the methods used to calculate pH50a and pH50ssd.

“ASIC currents were activated by changing the external solution from a conditioning pH of 7.4 -or any other indicated value- to a more acidic test pH for 5s to 30s depending on the desensitization rate of the channel tested. After activation and complete desensitization –return of the current to the zero value- or after the sustained current reached a plateau, the solution was changed to pH 7.4 or the indicated preconditioning pH for 15 to 30 s, short for non-desensitizing and long for desensitizing mutants. To calculate the pH50a each mutant channel was first activated with solutions in the pH range from 6.9 to 4.5. Lower pH values induced a sustained endogenous current in oocytes. Most channels reached Imax around pH 6.0; peak currents decreased at lower pH. Thus, we used a pH range from no activation to one that induced maximal current for the mutant tested. Channels with pH50a markedly shifted to the right, the range extended to pH 4.5.”

6) Generally, the Discussion section should place the work into a wider context. However, in its present form, the discussion lacks detail and should be expanded to include a more in-depth discussion on previous work (https://www.ncbi.nlm.nih.gov/pubmed/12947112; https://www.ncbi.nlm.nih.gov/pubmed/17389250), including comments or speculation on mechanisms of desensitization in non-mammalian ASICs (https://www.ncbi.nlm.nih.gov/pubmed/20064854).

We have modified the paragraph in the Discussion explaining why mutations in domains different from the Q-valve could affect desensitization and/or permit sustained currents as follows:

“Overall our results are consistent with previous reports of mutations and modifications of cysteine substitutions in the β1-β2 (Coric et al., 2003) and β11-β12 linkers (Springauf et al., 2011), in the β1 strand forming the lower palm (Cushman et al., 2007; Roy et al., 2013), TM1 (Li et al., 2010a) and even the thumb domain that makes contact with the upper part of TM1 (Krauson and Carattino, 2016). Recent structural and functional data have informed that those regions are the ones that undergo the largest conformational changes, and all are located between the Q-valve and the pore, consistent with transducing conformations to the gate. Hence, experimentally introduced mutations or evolution guided species-specific substitutions in any of those domains can alter desensitization and even enable sustained currents (as indicated in the kinetic schemes of Figure 2—figure supplement 1). This explains why not all channels that have the tried Q276-L415-N416 exhibit identical kinetics of desensitization.”